# Semiconducting Polymer-Based Nanocomposite for Photothermal Elimination of *Staphylococcus aureus* Biofilm

**DOI:** 10.3390/microorganisms13112568

**Published:** 2025-11-11

**Authors:** Pedro Sanchez, Erica Vargas, Stan Green, Madison Greer, Shaina Yates-Alston, Mariana Esposito, Li Tan, Nicole Levi

**Affiliations:** Plastic and Reconstructive Surgery, Atrium Health Wake Forest Baptist, Winston-Salem, NC 27157, USA

**Keywords:** photothermal therapy, near infrared light, nanocomposite, mild hyperthermia, medical devices, biofilms, *Staphylococcus aureus*

## Abstract

Biofilm growth on silicone (Si) medical devices is routinely treated with antibiotics or device removal; however, new approaches are needed. The current work evaluates photothermal therapy (PTT) to augment antibiotic efficacy or directly ablate *Staphylococcus aureus* biofilms. The semiconducting polymer, Poly [4,4-bis(2-ethylhexyl)-cyclopenta [2,1-b;3,4 b’]dithiophene-2,6-diyl-alt22,1,3-benzoselenadiazole-4,7-diyl] (PCPDTBSe), with a high photothermal conversion efficiency of 53.2%, was formulated into nanoparticles (BSe NPs) and incorporated into Si. Nanocomposites were stimulated with 800 nm light to generate mild hyperthermic conditions of 42 °C, or ablative temperatures above 50 °C. PTT, with or without antibiotics, was deployed against two strains of *Staphylococcus aureus* biofilms, Xen 29 and Xen 40, followed by an evaluation of bacterial survival, biofilm regrowth, and differential disruption of specific biofilm components. Mild hyperthermia was also used in an in vivo model of silicone implant infection. The results demonstrate a 55–59% reduction in *S. aureus* when PTT plus antibiotic was used in vitro, and a 51% reduction in vivo. Higher temperatures effectively eradicate both Xen 29 and Xen 40 strains, with a longer exposure time using lower laser power being optimal. Hyperthermia inhibited biofilm regrowth in both strains, resulting in a > 3 log reduction, plus increased dead cells, polysaccharides, and eDNA in treated Xen 40 biofilms. These experiments demonstrate that nanocomposite-based PTT can both reduce viable bacteria and alter individual biofilm components.

## 1. Introduction

Approximately 1.7 million patients develop healthcare-associated infections (HCAIs) each year, and more than 98,000 patients succumb to these infections [1]. The four types of HCAIs include central line-associated bloodstream infections (CLABSIs), catheter-associated urinary tract infections (CAUTIs), surgical site infections (SSIs), and ventilator-associated pneumonia (VAP) [1]. This includes temporary indwelling devices, such as urinary catheters, wound drains, and needleless connectors or implantation of medical devices, such as breast implants, cardiac pacemakers, and artificial joints [2,3]. Mortality rates are device-dependent, ranging from less than 5% for dental implants to greater than 25% for heart valves [4,5]. Both short- and long-term indwelling catheters usually comprise silicone, which is more easily infected than other materials [6,7]. Fifty to seventy percent of HCAIs are attributed to indwelling medical devices [1,8], and USD 28 to 45 billion is spent annually on HCAIs in the United States [9]. The use of antibiotics alone to eradicate biofilm-associated infections is becoming less effective, possibly due to poor antibiotic penetration throughout biofilms, as well as reduced metabolic activity and increased populations of persister cells [10,11].

*Staphylococcus aureus* is a Gram-positive commensal bacterium, with 30–40% of the human population being colonized, making it one of the most clinically significant pathogens in medical device-related infections [2,4,12,13]. *S. aureus* is responsible for 11–16% of all device-related infections, including catheter-related septicemia, prosthetic valve endocarditis, catheter-related bloodstream infections, septic arthritis of prosthetic joints, as well as most skin and surgical site infections [12,13,14,15,16]. The biofilm matrix produced by different *S. aureus* strains depends upon environmental factors, including shear, temperature, nutrient availability, and the presence of other microorganisms, which are crucial variables when considering alternative treatments to antibiotics [17,18,19,20].

Biofilm development consists of four stages which include attachment, micro-colony formation, maturation, and detachment/dispersal [21]. The first stage begins with planktonic bacteria attaching onto an abiotic or biotic surface and forming a monolayer [22,23]. As the bacterial cells proliferate, extracellular polysaccharide (EPS) is formed to protect the cells [23]. EPS is composed of lipids, proteins, polysaccharides, and extracellular DNA (eDNA,) which contribute to the biofilm structure [24,25]. Once planktonic bacteria irreversibly attach onto surfaces, they secrete the protective, highly hydrated EPS and proliferate into micro-communities, protected from host immune responses and antibiotics [25,26]. Biofilms have been associated with infections caused by drug-resistant Gram-positive bacterial species, such as *Staphylococcus aureus* [27], as well as Gram-negative bacteria, such as *Escherichia coli* and *Pseudomonas aeruginosa* [28]. Stewart [11] suggests that if antibiotics are inactivated, or sequestered, by binding to biofilm components, antimicrobial activity is reduced. *S. aureus* is susceptible to gentamicin in vitro, but it is not recommended as the primary source for treating bacterial infection [29] because the acquired resistance allows Staphylococcal strains to multiply in the presence of antibiotics [30]. Recalcitrant biofilms become resistant to high levels of antibiotics, leading to a higher chance of a recurring infection [31,32,33,34]. Therefore, antibiotics may not be the most effective form of treatment for biofilm-related infections [3]. Another option is removal of the medical device, although surgery may not be favorable to the patient, as it can be invasive and costly [3,35].

Hyperthermia is an elevated temperature induced locally or systemically, either inherently as the body’s immune response to an infection, or medically induced [36]. There are two classifications of hyperthermia treatment: mild hyperthermia and thermal ablation. Mild hyperthermia is defined as temperatures between 39 and 45 °C [37,38]. Thermal ablation includes temperatures above 45 °C, which induce irreversible protein denaturation and potentially tissue necrosis [39]. While mild hyperthermia (T < 45 °C) requires prolonged exposure for irreversible damage, temperatures above 60 °C exponentially reduce exposure time needed for destruction [40]. There are currently many studies that use hyperthermia to treat tumors [41,42,43]. In contrast, there are few studies demonstrating the effects of hyperthermia on biofilms, although this is a growing area, including a developing understanding of how heat can augment antibiotics [44,45,46,47,48]. In one study, *Staphylococcus epidermis* biofilm exposed to 60 °C for one hour induced significant reductions in bacterial viability and structural changes to the biofilm [44]. Hyperthermia has been demonstrated to disrupt bacterial membranes and protein synthesis [49,50,51,52]. For example, while most bacteria thrive between 33 °C and 41 °C, *E. coli*’s outer membrane can be disrupted at elevated temperatures (above 45 °C), leading to irreversible damage to bacterial structural integrity and disruption of cell proliferation [53].

Photothermal nanoparticles (NPs) can absorb light and convert it to heat, optimally within the near-infrared radiation (NIR) (700–900 nm) window, where water and hemoglobin have the lowest absorption [54]. Photothermal therapy (PTT) was first developed for oncologic applications using NIR photo-absorbers to generate heat for cell ablation at high temperatures or to augment chemotherapy [55]. Although PTT alone has demonstrated good potential for eradicating bacteria or biofilms [56], prolonged high temperatures (55–60 °C) may lead to tissue damage [55]. Therefore, the current work also explored an approach analogous to hyperthermia used for boosting chemotherapy, where 42 °C is known to be a safe temperature. Our team, and others, have shown that antimicrobial agents are enhanced using mild hyperthermia, and the current work uses a nanoparticle with high light-to-heat conversion. Silicone-based medical devices do not generate heat when exposed to laser light because of their poor ability to absorb near-infrared radiation. In order to remedy this, silicone was infused with poly [4,4-bis(2-ethylhexyl)-cyclopenta [2,1-b;3,4-b’]dithiophene-2,6-diyl-alt22,1,3-benzoselenadiazole-4,7-diyl] (PCPDTBSe) nanoparticles (BSe NPs) to create a silicone nanocomposite (BSe-Si) since these NPs have a documented ability to quickly generate heat with absorption of NIR light [57,58]. PCPDTBSe (Figure 1) is a novel semiconducting polymer, where the addition of selenium alters the band gap, and hence the optical absorption properties, based on the judicious choice of the polymer molecular weights [59]. Other photothermal nanoparticles, such as gold, silver, and carbon, have been used for PTT in nanocomposites, but to our knowledge, the current work is the first to use semiconducting polymer nanoparticles as the heat-generating component [60,61,62]. Semiconducting nanoparticles, or their free-floating nanocomposites, have been used against biofilm; however, nanocomposites as the device component without the release of nanoparticles or antimicrobial agents have not yet been shown [63,64]. In the development of nanocomposites, achieving uniform dispersion and mechanical integrity with prolonged use can be a limitation, especially when harder nanocrystals are utilized [65]. The use of a polymer-based PTT agent is expected to offer better interfacing with Si polymer chains, leading to reduced fatigue of the medical device with prolonged implantation time, or multiple heat–cool cycles. Inclusion of the PCPDTBSe NPs throughout Si is also expected to be an advantage since it eliminates the potential for delamination that can occur with thin-film nanoparticle coatings. There is also no need for the nanoparticles to be released from the composite to be effective, making them a durable resource for treating biofilms if infection should arise.

## 2. Materials and Methods

### 2.1. Disk Materials

PCPDTBSe polymers were synthesized using a microwave-assisted Stille coupling procedure of 4,4-bis(2-ethylhexyl)-2,6-bis(trimethylstannyl)-4Hcyclopenta [2,1-b;3,4-b0]dithiophene with 4,7-dibromo-2,1,3-benzoselenadiazole in the presence of a Pd(0) catalyst, similar to Coffin et al. [66]. For a detailed methodology of PCPDTBSe NP’s, the reader is encouraged to review MacNeill et al. [67]. BSe NPs were prepared similar to McCabe-Lankford et al. [68]. One milliliter of 2 mg/mL BSe polymer mixture in THF was injected into 8 mL of 1 mg/mL aqueous Pluronic F127 solution under constant horn sonication using a Branson Digital Sonifier (Brookfield, CT, USA) at 20% power for 1 min. Residual THF was evaporated, and the BSe NPs were then autoclaved at 250 °C for 30 min for sterilization. They were then centrifuged at 7000 rpm for 30 min, and the resulting supernatant was concentrated through further centrifugation at 16,800 rpm for 4 h before resuspension in water. Nanoparticles were characterized by dynamic light scattering (Malvern Panalytical Zetasizer Nano-ZS90, Malvern Panalytical, Malvern, UK) for size uniformity and zeta potential before incorporation into silicone. They were about 75 nm in diameter and −20 mV consistently amongst batches. They were maintained in a stable suspension in water for a prolonged time (months) before use, with no aggregation. They were also easily resuspended following centrifugation for concentration to the desired amounts prior to addition into silicone. The PCPDTBSe nanoparticle concentration was determined via optical absorption using a previously developed standard curve [68].

Both the Si and BSe-Si disks used a SYLGARD^TM^ Silicone Elastomer Kit as a base. To create the BSe-Si disks, a 10:1 ratio of base (Part A) and curing agent (Part B) was combined with the BSe NP. BSe NPs at a concentration of 120 mg/mL were added to Part A in a glass vial: 50 µL to 900 mg of Part A, followed by 100 mg of Part B (for 6 mg of BSe NPs per gram of silicone), or 417 µL to 4500 mg of Part A, followed by 500 mg of Part B (for 10 mg of BSe NPs per gram of silicone). The combination of Part A and BSe NPs was mixed until homogeneous, then sonicated for 15 min in an ultrasonic water bath (1510 BRANSON), spread into a 60 mm glass dish, and any remaining water evaporated by heating for 30 min at 120 °C. Then Part B was added, and the mixture was cured for 4 h at 45 °C. Si disks were created using the same method with the same 10:1 ratio for Parts A and B, but without the introduction of BSe NPs. PCPDTBSe nanoparticles were incorporated throughout the silicone to obtain a homogeneous mixture before silicone curing. PCPDTBSe nanoparticles were not modified in any way. There was no precipitation of the nanoparticles. Individual disks of Si and BSe-Si were obtained from the formed sheets using a 5 mm biopsy punch, and the disks were then sterilized by autoclave for 15 min at 273 °C.

### 2.2. Thermal Measurements

To evaluate the nanocomposite materials for the generation of mild hyperthermia, triplicate samples of Si or (6 mg/g) BSe-Si were placed into 96-well plates (CytoOne) with 200 µL of diH_2_O. A well with 200 µL of diH_2_O and no disk served as a control. Two laser treatment parameters were tested to achieve a ΔT of 5 °C, which would provide a treatment temperature of 42 °C by starting at the initial base temperature of 37 °C for the biofilm. Temperatures of the disks were measured by placing the tip of a fiberoptic probe (Qualitrol Neoptix^®^ with Nomad thermometer, Qualitrol Neoptix, Quebec City, QC, Canada) onto the disk before and after laser exposure. Thermal imaging of the Si and BSe-Si disks was also conducted using an FLIR thermal imaging camera. A K-Cube^®^ laser (Summus Medical Laser Inc., Franklin, TN, USA) was set to a 1 cm beam diameter and 800 nm continuous wavelength (CW). The laser used is an FDA-approved laser for dermal lesions and soft tissue pain management. Doses as high as 10 W can be safely applied for minutes at a time to human skin [69,70]. For mild hyperthermia, either 3 W for 25 s was applied (75 J), or else 5 W, 12 s (60 J). Laser parameters (time and power variations) were determined through trial and error to achieve a ΔT of 5 °C; then, triplicate samples were analyzed to ensure consistent heating among disks. To evaluate the potential for the BSe-Si disks to generate ablative temperatures, Si or BSe-Si disks were again placed in wells of a 96-well plate with 200 µL of diH_2_O and exposed to 1 W, 3 W, or 5 W of 800 nm light for up to 300 s, and the temperature increase, ΔT (°C), was recorded over time.

### 2.3. S. aureus Bacteria

This study evaluated two bioluminescent strains of *Staphylococcus aureus*, Xen 29 and Xen 40 purchased from Revvity, to evaluate the differences between strains that produce more or less exopolysaccharide. Bioluminescence is advantageous for identifying the metabolically active bacteria in vivo [71,72,73]. Xen 29 is from the parental strain, American Type Culture Collection (ATCC) 12600, derived from pleural fluid [74]. ATCC 12600 is known to produce high levels of EPS when developed under shear flow, and moderate amounts when grown statically [75,76,77]. Xen 40 is from the parental strain, UAMS-1, which is a clinical isolate from a patient with osteomyelitis; it has been described as a high exopolysaccharide producer [78,79,80]. Previous studies of UAMS-1 have shown that extracellular DNA and carbohydrates play a significant role in the biofilm structure and biomass fluctuations, with eDNA specifically being linked to facilitating microcolonies during dispersion [81,82,83]. Nutrient Broth No. 1 (NB1) and NB1 agar plates were used for Xen 29, and Tryptic Soy Broth (TSB) and TSB agar plates were used for Xen 40.

### 2.4. Biofilm Development

To encourage growth of the biofilm, the disks (either 6 mg/g of BSe NPs for mild hyperthermia or 10 mg/g of BSe NPs for ablative evaluation) were coated with human plasma. Wells of either 96, 48, or 24 well plates were filled with 0.2 to 1 mL of 20% human plasma diluted in 1 M sodium bicarbonate, incubated at 4 °C overnight, and then the plasma was aspirated, and the plates were dried for 1 hr under sterile air flow before the addition of bacteria. A single colony of *S. aureus* grown on the corresponding nutrient agar plate was inoculated into 30 mL of broth and placed in a shaker incubator at 37 °C and 150 rpm for 24 h. Bacteria were then diluted to an OD of 0.2 (mild hyperthermia experiments) or 0.8 (for ablative experiments) at 600 nm (OD_600_) and added at 0.2, 0.4, or 1 mL for the use of 96, 48, or 24 well plates, respectively. Plate lids were lined with sterile aluminum foil to reduce oxygen tension to promote biofilm growth, and sealed plates were placed in a static incubator for 24 or 48 h at 37 °C to allow biofilm development on the disks. Biofilms were then washed two times with 1x PBS and prepared for mild hyperthermia, luminescence analysis, or ablation, as described below.

Bioluminescence was performed on disks to confirm the presence of biofilms before implantation into mice. After washing the biofilms, all the PBS was removed from each well, and 200 µL of fresh NB1 broth was added to each well. The plates were then imaged for 5 min using a Perkin Elmer in vivo imaging system (IVIS) (PerkinElmer, Waltham, MA, USA), and samples were quantified by measuring the regions of interest (ROIs) of photons within the whole-well diameter of each biofilm sample. This analysis confirmed the presence of biofilms prior to sonicating and vortexing the samples for quantification via spectrophotometry.

### 2.5. Photothermal Therapy

#### 2.5.1. Mild Hyperthermia with Antibiotics In Vitro

Fifty milligrams of powdered gentamicin (SIGMA-ALDRICH) was added to 10 mL of diH_2_O, and a diluted stock of 1 mg/mL was prepared from the 50 mg/mL solution. The 1 mg/mL stock was further diluted to a working stock of 0.0867 mg/mL, 3 times the Xen 29 MIC of 0.0289 mg/mL, and stored at 4 °C. Xen 29 biofilms were prepared as described above in 96-well plates, with triplicate samples per group, with 24 h incubation for biofilm development. Following washing to remove planktonic bacteria, 133.4 µL of NB1 broth was added to each well, with either 66.6 µL more NB1 (control) or gentamicin (0.0867 mg/mL). All samples were then incubated at 37 °C for 1 h before exposure to the laser. Treatment groups included biofilm on Si or (6 mg/g) BSe-Si disks with gentamicin and no laser (control), laser only (3 W for 25 s or 5 W for 12 s), or laser and gentamicin (3 W for 25 s or 5 W for 12 s). During laser exposure, the plate was maintained at 37 °C on a heat block. After laser exposure, the plates were returned to 37 °C for an additional 3 h to allow time for antibiotic diffusion. Then, all of the liquid (NB1, gentamicin, and planktonic bacteria) was aspirated, residual biofilms were washed twice with PBS, 200 µL of NB1 was added, and the samples were imaged using IVIS to capture the bioluminescence of the biofilms. Photon flux was quantified over each well (region of interest (ROI)). All groups were evaluated with triplicate samples. Following IVIS imaging, each disk was sterilely transferred to a 1.5 mL centrifuge tube containing 200 µL of NB1 broth. Tubes were vortexed for 30 s and sonicated in an ultrasonic water bath to disrupt biofilms. After sonication, the tubes were vortexed for 10 s, followed by serial dilution and plating onto NB1 agar. Agar plates were incubated at 37 °C for 16 h, and colony-forming units (CFUs) were counted.

#### 2.5.2. Mild Hyperthermia with Antibiotics In Vivo

For in vivo evaluation of mild hyperthermia with gentamicin, Si and (6 mg/g) BSe-Si disks were autoclaved, and Xen 29 biofilms were grown in 24-well plates, as described previously, with 24 h for biofilm development. IVIS imaging was performed to confirm that the disks had metabolically active bacteria before implantation into mice. Animal usage was approved by the institutional IACUC, and animals were handled humanely. Since mouse skin is thin, before we initiated the study, we tested 3 W, 25 s and 5 W, 12 s (these parameters generate the same amount of heat (about 5 °C) from the BSe-Si disks) on the skin of hairless mice without the inclusion of BSe-Si disks. We observed dermal damage with 5 W, 12 s but not with 3 W, 25 s. A total of 24 Crl:SKH1-hrBR hairless immune-competent mice were used in the PTT experiments. Mice were anesthetized with inhalation of 2% isofluorane with supplemental oxygen. Skin was cleansed with 70% ethanol, 2% betadine, and 70% ethanol a second time. Following drying of the alcohol, a 1 cm incision was made in each flank. The skin was elevated, and the disks with biofilm were placed onto the underlying fascia and secured to the skin with one stitch, followed by closure of the incision with two more stitches. All mice received one 5 mm Si disk implanted subcutaneously in the left flank and one 5 mm (6 mg/g) BSe-Si disk implanted subcutaneously in the right flank. Biofilm-laden disks were placed 24 h prior to PTT, and IVIS imaging was performed 1 h before PTT to obtain a baseline of bioluminescence to aid in quantifying bacterial burden. Mice received an intraperitoneal injection of either 200 µL of 2 mg/mL of gentamicin or saline, 1 h before laser exposure, followed by laser exposure (3 W, CW, 800 nm, 25 s) applied directly over the disk in the flank. Although the in vitro photothermal evaluation demonstrated that 800 nm light delivered at 3 W for 25 s or 5 W for 12 s could both generate a ΔT of 5 °C to provide the mild hyperthermic temperature of 42 °C, assuming the core body temperature of 37 °C; preliminary testing of these laser parameters on mouse skin demonstrated that 5 W could cause dermal burning. Therefore, only 3 W of 800 nm light delivered for 25 s was deemed safe for mild hyperthermia and utilized in animals. One animal received the biofilm-laden disks and received saline injections only. Five mice received biofilm-laden disks and were treated with systemic gentamicin only. Seven mice received biofilm-laden disks and were treated with systemic saline injection and 3 W of 800 nm light for 25 s, applied directly over each flank. Eleven mice received biofilm-laden disks and were treated with systemic gentamicin plus 3 W of 800 nm light for 25 s, applied directly over each flank. Twenty-four hours after treatment, mice were imaged again with IVIS to determine Xen 29 bioluminescence after treatment. Following euthanasia by carbon dioxide asphyxiation, the disks and surrounding tissue were extracted, weighed, and homogenized. The homogenate was diluted up to 10^−6^ and plated in triplicate on NB1 agar to calculate the bacterial burden per disk, defined as colony-forming units per gram of homogenate (CFU/g).

#### 2.5.3. Ablative Hyperthermia In Vitro

Laser parameters were the same as for the pre-biofilm trials (1 cm beam diameter, 800 nm CW) and either 1 W, 3 W, or 5 W of power applied for 300 s, 100 s, or 60 s, respectively. Each trial was performed in triplicate, with 3 Si and 3 BSe-Si disks. Biofilms of either Xen 29 or Xen 40 were developed on disks in 48-well plates, as described above. Following biofilm growth for 48 h, sterile pick-ups were used to extract each disk, which was then dunked in sterile water and transferred to a new 96-well plate. Wells were then filled with 200 µL of water and placed in an incubator for 15 min at 37 °C to achieve a standardized initial temperature. Plates were kept at 37 °C during laser exposure by placing them on a heating block. After laser exposure, plates were returned to the 37 °C incubator for 15 min before being sealed in parafilm and sonicated in an ultrasonic water bath for 2 min to disrupt the biofilms. Each well was then mixed using a pipettor, serially diluted, plated onto NB1 agar, and incubated at 37 °C overnight, followed by CFU enumeration.

A potential benefit of using a nanocomposite is rapid hyperthermia; hence, photothermal heating versus rapid heat shock via classically established methodology was compared. Xen 29 or Xen 40 biofilms were exposed to hyperthermia using a circulating hot water bath (PolyScience MX immersion circulator, Polyscience, Niles, IL, USA). PTT analysis of heat generated by the BSe-Si disks in 200 µL volume of water revealed that 3 W applied for 100 s led to a ΔT of 26.5 °C, and 5 W applied for 60 s led to a ΔT of 28.3 °C. Based on these values, we utilized a maximum of ΔT of 27 °C for PTT using either laser parameters; hence, 27 °C + 37 °C = 64 °C. However, this was the maximum temperature change achieved and may not be representative of the average temperature change that the biofilms experience as the temperature ramps to its maximum over time. Therefore, we also utilized the approximate average between 37 °C and 64 °C or 50 °C to evaluate the average temperature that the *S. aureus* biofilms experience during PTT. Heat shock evaluation did not need to involve the use of the nanocomposite materials, as the goal was to understand susceptibility of the biofilms to these temperatures. Xen 29 or Xen 40 biofilms were grown in sterile 1 mL Eppendorf tubes that had previously been coated with human plasma, as described above. Tubes were filled with either 200 µL of 0.2 OD_600_ Xen 29 or Xen 40 and then placed in a static incubator at 37 °C for 24 h to develop biofilms. Biofilms were then washed with sterile water, 200 µL of water was added to the tubes, and the biofilms were submerged in hot water for 60 s (analogous to using 5 W, 60 s ablation) or 100 s (analogous to using 3 W, 100 s ablation) at either the average temperature of 50 °C or the maximum temperature of 64 °C. Following heat shock, the tubes were sonicated in an ultrasonic water bath, and samples were utilized for bacterial viability quantification using serial dilution, plating on agar and CFU enumeration. Alternative samples exposed to heat shock were evaluated from biomass development analysis by immediately removing the water and fixing the biofilm with ice-cold methanol. Biofilms were then stained by adding 300 µL crystal violet (CV) (0.06% crystal violet in water) for 10 min, rinsed with water, and air-dried. CV was eluted by shaking with 200 µL of elution buffer (5.88 g of sodium citrate, 50 mL of ethanol, and 150 mL of water) for 10 min, then optical absorption was read at 590 nm using a Tecan Infinite 200 Pro plate reader. CV absorption is a metric of total biomass, and each treated sample was normalized to the unheated controls. The potential for viable bacteria to regrow biofilm was evaluated by inoculating wells of a 24-well plate that had been pre-coated with human plasma and filled with 1.9 mL of NB1 broth with 100 µL of the biofilm homogenate that had been exposed to hyperthermia. Plates were incubated at 37 °C for 48 h and then quantified by CV staining for biomass analysis, with samples being normalized to controls that regrew biofilm with no hyperthermia (37 °C only). All samples were performed in triplicate.

### 2.6. Fluorescence Imaging of Biofilms

Xen 29 and Xen 40 biofilms were developed for 48 h on Si or 10 mg/g BSe-Si disks and exposed to 800 nm of laser light at 1 W for 300 s, 3 W for 100 s, or 5 W for 60 s, with control disks not exposed to laser. Following PTT, disks were stained with SYTO 9 to quantify live bacteria, propidium iodide (PI) to quantify dead bacteria, wheat germ agglutinin 555 (WGA) to measure polysaccharide content of the biofilms, and TOTO-3 iodide to quantify extracellular DNA in the biofilms. Following PTT, Si and BSe-Si disks were immediately moved to clean well plates, stained according to the manufacturer’s guidelines, and imaged using a Keyence BZ-X 800 microscope (Keyence Corporation, Osaka, Japan), using appropriate filter cubes for Syto 9 (485 excitation/498 emission), PI (535 excitation/617 emission), WGA (555 excitation/565 emission), and TOTO-3 (642 excitation/660 emission). Fluorescence intensity from the images was captured using Photoshop for statistical comparison.

### 2.7. Statistical Analysis

A one-way analysis of variance (ANOVA) was performed to determine if groups were statistically significant. However, there were a few groups that had a small range of variances and required a Kruskal–Wallis test in Microsoft Excel to determine their statistical significance. A *p*-value of less than 0.05 was considered significant, and data is represented as the mean ± the standard error of the mean.

## 3. Results

### 3.1. Temperature Changes in BSe-Si for Mild Hyperthermia

Heat generation of Si and BSe-Si disks was evaluated with exposure to a continuous wave (CW) laser at 800 nm under two power conditions: 3 W for 25 s and 5 W for 12 s. As shown in Figure 1, temperature increases in Si alone were analogous with water, indicating that Si exposed to laser does not generate heat. BSe-Si had temperature increases of 5.07 and 5.03 °C, while Si only had an increase of 0.83 at 3 W or 2.27 at 5 W. Differences in heat generation between Si and BSe-Si were statistically significant, confirming that BSe-Si confers PTT capability. The result that both 3 W and 5 W had about a ΔT = 5 °C demonstrates that the total laser energy (75 J for 3 W or 60 J for 5 W) may not be as impactful as the time of exposure.

### 3.2. Xen 29 Biofilm Reduction with Mild Hyperthermia and Gentamicin

As shown in Figure 2A, when evaluating the bioluminescence of the Xen 29 biofilms grown on Si or BSe-Si, there was no statistically significant difference in the number of emitted photons, regardless of treatment with 3 W or 5 W of 800 nm laser light, with or without gentamicin. The overall responses are comparable, suggesting that the slight temperature changes do not make an appreciable difference in bacterial viability. The result also shows that the addition of the antibiotic to PTT does not reduce viable bacteria compared to simply using the antibiotic alone. Although the bioluminescence intensity results did not indicate differences in metabolically active bacteria immediately after treatment, disruption of the Xen 29 biofilm and quantification of CFUs did showcase a benefit of the nanocomposite-based PTT. As shown in Figure 2B, application of 3 W or 5 W without gentamicin resulted in significant reductions in bacteria, with Si exposed to 3 W having a 53% reduction and exposure to 5 W having a 71% reduction, whereas BSe-Si had a 58% and 55% reduction with 3 W or 5 W exposure, respectively. Although statistically different, all the reductions in CFUs were less than 1 log, which is not clinically significant (where greater than 3 log is needed). An addition of gentamicin did not augment CFU reductions for either Si or BSe-Si exposed to 3 W for 25 s, although gentamicin did provide a 63% reduction for Si and 59% for BSe-Si treated with 5 W for 12 s.

### 3.3. Biofilm Reduction In Vivo

Prior to implantation, Xen 29 biofilm growth on Si and BSe-Si disks was confirmed by bioluminescence imaging, as shown in Appendix A, where ample luminescence on either material confirmed the presence of metabolically active bacterial biofilms. Bioluminescence from biofilm-coated disks following implantation in mice, before (Appendix A), as well as 24 h after (Appendix A), mild hyperthermia plus the antibiotic gentamicin was also observed, demonstrating a reduction in metabolically active bacteria 24 h after treatment. The results support the utility of using bioluminescent bacteria as a tool for in situ evaluation of PTT. Appendix A shows that bulges of the disks under the skin can be observed, although the green color of BSe-Si was not apparent. Blanching of the skin over the implants and redness were observed before PTT, suggesting infection. Immediately following laser exposure (3 W for 25 s), skin blanching was observed over the regions for both the Si and BSe-Si disks, although the BSe-Si disk had profoundly more. About 24 h after PTT, the skin above the disk area was scabbed, suggesting initial healing, although whether the healing was from thermal injury or infection was unable to be determined since the same observations were made for animals treated with no laser also.

To assess the efficacy of PTT combined with systemic gentamicin in vivo, colony-forming units per gram of tissue (CFU/g) were quantified from homogenized *S. aureus* Xen 29 biofilms grown on either Si or BSe-Si disks. Mice received one of four treatments: no laser/no gentamicin (NL/NG), laser only (NaCl + L), gentamicin only and no laser (NL + G), or combined gentamicin and laser (G + L). As shown in Figure 3A, the lowest CFU counts were observed in the G + L BSe-Si group, with a 51% reduction, suggesting enhanced bacterial reduction for PTT used in conjunction with an antibiotic. Although a statistically significant reduction in CFUs was measured, this was less than 1 log, which is not clinically consequential. Nonetheless, these findings support the potential of BSe-Si-based PTT to augment systemic antibiotics for reducing biofilm-associated infections on implanted polymeric materials.

To quantify the impact of PTT on metabolically active *S. aureus* biofilms, bioluminescence imaging was performed before and after treatment, and the percent change in photon flux was calculated in regions of interest (ROIs) completely encompassing the implantation sites. As shown in Figure 3B, the greatest reduction in photon flux was observed in the G + L BSe-Si group (−26%), followed by G + L Si (−13%), indicating reduced bacterial viability when PTT was combined with systemic gentamicin. There was a minimal reduction (4%) for BSe-Si with no laser and gentamicin, yet Si had an 18% increase in photon flux. There was also increased bioluminescence for disks in animals treated with Si (NL + G), or Si and BSe-Si treated with saline plus laser. The results indicate that laser alone on Si implants augments antibiotics and support that the heat generated by BSe in Si may improve the outcome by more than doubling the reduction in viable bacteria based on bioluminescence quantification. The magnitude of bioluminescence reduction in the G + L BSe-Si group supports the potential of nanocomposite-mediated PTT to enhance bacterial clearance in vivo.

### 3.4. Temperature Changes in BSe-Si with Variable Laser Parameters for Ablative Hyperthermia

Although the results for the in vivo analysis were promising, the initial in vitro and in vivo results indicate that an alternative BSe-Si nanocomposite formulation may be better. Therefore, we explored using a 1% loading of the NPS, which led to a uniform green color indicative of homogeneous NP dispersion, and these composites were used in all subsequent experiments. Temperature changes were recorded for Si and BSe-Si disks exposed to 800 nm of laser light at 1 W, 3 W, and 5 W for up to 300 s to determine the photothermal efficiency of the nanocomposite material. BSe-Si disks exhibited significantly greater temperature increases compared to Si across all power levels (Figure 4 and Appendix A). Beginning from a start temperature of 37 °C and adding the measured temperature changes as shown in Figure 4, regardless of laser power, none of the Si samples could achieve ΔT > 10 °C, or over 47 °C, which is insufficient for killing bacteria. BSe-Si generates much higher temperatures, with 1 W, 300 s reaching ΔT = 18 °C, and 3 W, 100 s having ΔT above 50 °C. We could not measure a ΔT at 5 W, 300 s due to boiling of the water around the disk and inaccuracies in temperature measurement, although BSe-Si exposed to 5 W for 90 s has a ΔT = 48 °C, which is more than sufficient to ablate bacteria (37 + 48 = 85 °C). Appendix A demonstrates Si and BSe-Si disks evaluated at a starting room temperature of about 23 °C. Si exhibited maximum temperatures of 24.2, 23.6, and 27.3 °C, with 1 W, 300 s; 3 W, 100 s; or 5 W, 60 s of laser irradiation, which shows that it is incapable of serving as a photothermal compound on its own. BSe-Si disks demonstrated uniform temperature generation over the entirety of the nanocomposite, supporting homogeneous distribution of the PCPDTBSe nanoparticles. Maximum temperatures of 43.1, 50.8, and 49.9 °C for 1 W, 300 s; 3 W, 100 s; or 5 W, 60 s demonstrate the photothermal potential and confirm that either 3 W, 100 s or 5 W, 60 s generate the same amount of heat. Figure 4 shows that BSe NPs conferred heat generation potential to silicone yet also demonstrates that lower laser powers require longer times to achieve higher temperatures. These results confirm the superior photothermal conversion capability of BSe-Si, enabling precise thermal modulation for either mild hyperthermia or ablative therapy, depending on the applied laser power.

### 3.5. PTT for Ablative Hyperthermia

As shown in Figure 5A, Xen 29 CFU enumeration revealed a progressive reduction in bacterial viability with increasing laser power in the BSe-Si groups, with 81% and 83% reductions for 1 W, 300 s and 5 W, 60 s, respectively. Even though laser fluence remained the same, at 300 J for all groups, the 3 W, 100 s treatment led to complete eradication of Xen 29 bacteria. In contrast, Si disks showed minimal reductions (less than 1 log), indicating that both material type and laser parameters influence bacterial viability, with the BSe-Si nanocomposite offering great potential to ablate *S. aureus* bacteria compared to unmodified silicone. The impact of ablative PTT on Xen 40 biofilms (Figure 5B) reveals again that BSe-Si reduces bacterial viability with increasing laser power. Both 3 W, 100 s and 5 W, 60 s completely eliminated viable Xen 40 bacteria, whereas 1 W, 300 s had a 99.7% (2.5 log) reduction. Comparison between Xen 29 and Xen 40 indicates that Xen 40 is more susceptible to PTT, although 3 W, 100 s confers the same elimination of bacteria for both strains. Although all PTT treatment groups utilize the same laser fluence of 300 J, 1 W, 300 s is not as effective compared to higher laser powers applied for shorter times.

### 3.6. Biofilm Response to Heat Shock

In this work, we sought to standardize PTT using the same energy fluence since 3 W, applied for 100 s, and 5 W, applied for 60 s, both generate the same ΔT = 27 °C. Using a starting temperature of 37 °C, we concluded that the maximum temperature attained in a 200 µL volume of water around the disks could be 27 + 37= 64 °C, which should be sufficient for bacterial ablation. This temperature was applied to biofilms (using a hot water bath) for 60 s or 100 s to begin to understand if PTT would be more beneficial than bulk heating. Since it takes time to achieve this maximum temperature of 64 °C, we also evaluated the average temperature that the biofilms would experience with PTT (the average of 37 and 64 °C being 50 °C), which was applied to biofilms using the water bath. As shown in Figure 6A,B, CFU enumeration of Xen 29 and Xen 40 biofilms revealed progressive reductions in bacterial viability with increasing temperature and exposure time, with the greatest reduction observed at 64 °C for 100 s. Reductions in CFUs suggest that elevated temperatures, particularly at 64 °C, may effectively compromise biofilm viability. Notably, PTT (Figure 5) can completely eliminate viable bacteria, whereas the use of bulk heating, while able to reduce bacteria, is not as effective. It might be argued that there could be differences in biofilms grown on silicone (Figure 5) versus polystyrene tubes (Figure 6); however, controls for both cohorts have approximately the same number of bacteria, 1E8 CFU/mL. Therefore, the results indicate a benefit of BSe-Si-induced PTT, most likely due to the close approximation of the biofilm bacteria via direct thermal transfer from the nanocomposite to the biofilm, versus the indirect heating from the water to the biofilm that occurs in the heat shock experiments. PTT has the capacity to completely eliminate biofilm bacteria for both Xen 29 and Xen 40, while bulk heating of the biofilm only results in a 2.3 to 3.3 log reduction for Xen 29 or a 2.7 to 5 log reduction for Xen 40. Of note are the differences between Xen 29 and Xen 40 with bulk heating, as shown in Figure 6. While the prolonged time of 100 s at a lower temperature of 50 °C provides a 2.3 log reduction in Xen 29 (99.8% reduction), and a one log reduction when 60 s is used, Xen 40 exhibits no statistically significant loss of CFUs using bulk heating to 50 °C for either length of time. Using the maximum temperature increase attained with PTT (64 °C), Xen 29 had 3 log (99.95%) and 3.3 log (99.99%) reductions in CFUs with 60 s and 100 s exposure. At 60 s exposure to 64 °C, Xen 40 had a 2.7 log (99.6%) reduction but exhibited a superior reduction (5 log (100%)) when 64 °C was applied for 100 s. This data confirms that strain differences in response to hyperthermia may determine overall clinical utility.

Figure 6C,D show the quantified Xen 29 and Xen 40 biofilm mass following water bath hyperthermia (bulk heating) using crystal violet staining, normalized to the control. The same progressive reduction in biomass was observed with increasing temperature and exposure time as was seen in the CFU reductions, with the lowest biomass detected at 64 °C for 100 s. As seen in Figure 6C, for Xen 29 treated at 50 °C, there was a 29% reduction in biomass when exposed for 60 s, which is comparable to a 36% reduction at the 100 s exposure. Treatment of Xen 29 biofilms at 64 °C for 60 s was also comparable, with a 30% loss. However, biomass was reduced by 72% when exposed to 64 °C for 100 s. As seen in Figure 6D, although Xen 40 treated with 50 °C for 60 s had a 22% decrease in biomass, which is somewhat comparable to the response of Xen 29, there was a 73% reduction when exposed for 100 s. Xen 40 treated with 64 °C had about the same reduction in biomass when exposed for either 60 s or 100 s (81 versus 84%).

Application of heat shock as a non-antibiotic strategy for biofilm disruption demonstrated significant effects on both bacterial viability and biofilm architecture. Exposure of *S. aureus* Xen 29 and Xen 40 biofilms to elevated temperatures (50 °C and 64 °C) for short durations (60 or 100 s) resulted in marked reductions in CFUs and biofilm biomass. These findings suggest that heat shock may compromise biofilm integrity through mechanisms such as protein denaturation, membrane destabilization, and degradation of the extracellular matrix, particularly at higher temperatures and longer exposure times. There were unexpected differences between Xen 29 and Xen 40. Although Xen 29 had a reduced number of viable bacteria and biomass at 50 °C, Xen 40 did not have a loss of bacteria, although prolonged time led to biomass reductions, with 100 s being more profound. At the higher temperature of 64 °C, prolonged exposure to hyperthermia did not kill more Xen 29 bacteria, but it did kill more Xen 40 bacteria. Elevated hyperthermia needed a longer time against Xen 29 to reduce biomass, although Xen 40 was quite susceptible. Importantly, the consistent downward trend in both CFU counts and biomass supports the hypothesis that thermal stress can effectively impair *S. aureus* biofilms. These results align with previous studies indicating that biofilms are susceptible to thermal disruption and further highlight the potential of heat-based therapies as adjuncts to conventional antimicrobial approaches, especially in the context of device-associated infections, although sensitivity of the bacterial strains to hyperthermia appears to be an important factor in efficacy.

### 3.7. Biofilm Regrowth Following Heat Shock

To assess the impact of thermal stress on biofilm regrowth, Xen 29 and Xen 40 biofilms were exposed to heat shock at 50 °C or 64 °C for 60 or 100 s, followed by a 48 h incubation period at 37 °C to evaluate biomass recovery via crystal violet staining and normalized to the control maintained at 37 °C. As shown in Figure 7A,B, Xen 29 and Xen 40 treated with bulk heating, using a hot water bath and examined for biofilm regrowth, confirm that for both strains of *S. aureus,* longer times at lower temperatures (50 °C) are beneficial for reducing biomass. Interestingly though, 100 s of exposure to 50 °C led to a 63.7% reduction in biomass for Xen 40, compared to only a 34.3% reduction in Xen 29. As expected, elevated temperatures, 60 s or 100 s at 64 °C, provided greater benefit for reducing biomass regrowth, with 86.2 and 88.5% reductions for Xen 29. The lower time of exposure (60 s) at 64 °C for Xen 40 was not as effective, with only a 53.3% reduction, although longer exposure (100 s) resulted in an 85.3% reduction, which was analogous to Xen 29. These findings demonstrate that elevated temperatures not only reduce existing biofilm biomass but also impair the ability of *S. aureus* to re-establish biofilms post-treatment. Although there were differences between the two *S. aureus* strains, both were susceptible to elevated temperatures to ensure the minimal amount of regrowth. It was not expected that 15% biofilm mass would regrow, making it further critical to understand what biofilm components remain following exposure to hyperthermia. While the PTT results indicated no viable CFUs, the heat shock regrowth experiments are useful for interpreting that biofilm regrowth remains a possibility and must be considered, which is a future evaluation that must be conducted following PTT.

### 3.8. Microscopic Variance in Biofilms After PTT

Fluorescence imaging was used to assess the structural and cellular changes in Xen 29 and Xen 40 biofilms following PTT on Si or BSe-Si disks, with images in Figure 8 being representative, as all groups were evaluated in triplicate to determine statistical significances. Biofilms were stained with SYTO 9 for live cells, propidium iodide (PI) for dead cells, wheat germ agglutinin (WGA) for polysaccharides, and TOTO-3 for extracellular DNA (eDNA). Across all laser powers (1 W for 300 s, 3 W for 100 s, and 5 W for 60 s), BSe-Si disks exhibited greater disruption than Si disks, with increased PI (red fluorescence) indicating elevated bacterial death. Additionally, enhanced WGA (blue) and TOTO3 (yellow) signals reflected increased polysaccharide and eDNA content, which was especially notable for Xen 40, suggestive of matrix changes. There was variability in polysaccharide staining, although Xen 29 showed a downward trend following PTT using any laser parameters. At 1 W, 300 s Xen 40 had decreased polysaccharides, 3 W, 100 s had no alteration, and yet 5 W, 60 s exhibited a drastic increase. These results confirm that PTT not only reduces bacterial viability but also alters the biofilm composition, supporting its potential as a multifaceted strategy for biofilm eradication.

Appendix A presents changes in fluorescence intensity corresponding to the microscopy images in Figure 8. Across both strains, SYTO 9 fluorescence decreased significantly with increasing laser power, indicating reduced bacterial viability. Conversely, PI fluorescence increased, confirming enhanced bacterial death. WGA and TOTO-3 signals also rose post-treatment, suggesting quantifiable alterations in biofilm, especially an increase in eDNA for Xen 40, which was also observed in the control BSe-Si disk without laser exposure, suggesting the potential for BSe NPs to disrupt biofilm without hyperthermia, possibly due to the semiconductive nature of the BSe NPs that can alter electrical cell–cell signaling.

A notable observation between Si and BSe-Si that occurs for both Xen 29 and Xen 40 was the unexpected reduction (60% for Xen 29 and 44% for Xen 40) in viable bacteria on BSe-Si versus Si. As shown in Appendix A, Xen 29 had 98, 92, and 97% decreases in viable bacteria on BSe-Si compared to Si treated with 1 W, 300 s; 3 W, 100 s; or 5 W, 60 s, respectively. For the same parameters, Xen 40 on BSe-Si had 91, 91, and 74% reductions. Although variations in viable bacteria may be seen in Appendix A, a minimal reduction in viable bacteria was observed for Xen 40 treated with 5 W, 60 s, which also had the lowest increase in the number of dead cells (Appendix A). This was an unexpected result, since 3 W, 100 s and 5 W, 60 s have the same total energy fluence of 300 J and achieve the same temperature change, indicating that longer times for PTT may be more impactful than the temperature alone. Residual polysaccharides and proteins can provide a supporting template for biofilm regrowth. As shown in Appendix A, Xen 29 had a reduction in polysaccharides for all PTT treatments (2697, 200, and 206% for 1 W, 300 s; 3 W, 100 s; and 5 W, 60 s). Xen 40 had a 607% decrease for 1 W, 300 s, and no change for 3 W, 100 s, but had a profound increase for 5 W, 60 s (461%). Increased polysaccharide was not expected for ablative hyperthermia since it was anticipated to facilitate removal of the biofilm with washing. The statistically significant increase in polysaccharides for Xen 40 when 5 W, 60 s was used may indicate that the rapid treatment induced aggregation of polysaccharides such that they were more difficult to remove. This data is not supportive of increased polysaccharide production since the biofilms were not allowed to regrow but is indicative of components that may be more easily removed with biofilm washing. The most profound difference between Xen 29 and Xen 40 biofilms with differing PTT was observed in changes to eDNA. Extracellular DNA is expected to be released into the biofilm when bacteria are damaged by PTT. Xen 29 on BSe-Si had no change in eDNA between BSe-Si and Si, with no laser exposure or 1 W, 300 s, but had a 41% decrease for 3 W, 100 s and a 228% increase for 5 W, 60 s. Appendix A shows significant elevation of eDNA for Xen 40 across all PTT regimes, and also for BSe-Si with no laser exposure, with a 191% increase in eDNA compared to 163, 232, and 251% increases in eDNA for 1 W, 300 s; 3 W,100 s; and 5 W, 60 s. This result supports the data in Figure 7, which shows that Xen 40 had higher biofilm regrowth compared to Xen 29, possibly due to the increased presence of eDNA and polysaccharides, which support the biofilm structure.

## 4. Discussion

Nanocomposites may be much better than free-floating nanoparticles for their ease of use and the direct thermal transfer to the biofilm for bacterial killing and inhibition of biofilm regrowth. Although other polymers, such as polydopamine, can be used due to their biocompatibility, their temperature increases are not optimal, necessitating the development of polymers with better photothermal conversion efficiency [64,84,85,86,87,88,89,90]. Given that the laser light will penetrate the entirety of the thickness of the BSe-Si disk (1 mm) for heat generation, quantifying the number of nanoparticles at the surface was not performed since heat from all nanoparticles would contribute to overall heating. Silver and copper were developed into thin-film composites to inhibit biofilm development, mainly through the release of metallic ions, which can be compromised with native protein opsonization that impedes antimicrobial efficacy [91,92]. Our goal with developing the nanocomposite in lieu of a thin-film coating on silicone was to eliminate the potential for thin-film delamination, plus being able to generate more heat by infusing the entirety of the silicone with a low loading of photothermal nanoparticles. There is also no need for the nanoparticles to be released from the composite to be effective, making them a durable resource for treating biofilms if infection should arise. Our results also show that heat can reduce biofilm regrowth, which makes it especially valuable for eliminating resistant infections, although the development of heat resistance and proof of efficacy of PCPDTBSe nanocomposite silicone needs to be evaluated against a broad spectrum of bacteria.

Although mechanical properties were not measured, the low loading of PCPDTBSe NPs is unlikely to alter mechanical integrity if the composites are implanted for a prolonged time. A major feature of using the PCPDTBSe nanocomposites is developing a more robust framework for how PTT-based heating adjacent to the biofilm may kill bacteria, plus alter biofilm composition and regrowth. The current study demonstrates the efficacy of PTT using PCPDTBSe nanoparticle-infused silicone (BSe-Si) for the disruption and eradication of *S. aureus* biofilms, both in vitro and in vivo. The superior performance of BSe-Si over unmodified silicone across all laser powers highlights that a low loading of NPS (1% or less) is needed, and laser treatment parameters can be altered to vary the amount of heat generated. During the course of these experiments, Si and BSs-Si disks were autoclaved many times between experiments, with no damage or loss of PTT potential for BSe-Si, supporting their utilization as implantable effective anti-biofilm strategies. The results support the hypothesis that nanocomposite-mediated hyperthermia can enhance antibiotic efficacy and directly ablate biofilm structures, offering a promising alternative to conventional antibiotics alone for device-associated infections.

Only the Xen29 strain was used in vivo, and there may be differences in *S. aureus* strains’ responses to hyperthermia. Hence, we selected to evaluate the moderate biofilm-producing strain of Xen 29 for comparison against Xen 40, which has been described as producing copious biofilm. This choice also allowed us to account for any strain differences in biofilm adhesion to silicone, where we found that Xen 29 had better adhesion to silicone. Histopathology was not performed since the disks needed to be removed with the surrounding tissue and evaluated for bacteria. Since mouse skin is thin, before we initiated the study, we tested 3 W, 25 s and 5 W, 12 s (these parameters generate the same amount of heat (about 5 °C) from the BSe-Si disks) on the skin of mice without the inclusion of BSe-Si disks. We observed dermal damage with 5 W, 12 s but not with 3 W, 25 s. Hence, we used 3 W for treating the mice with BSe-Si disks. Mild hyperthermia (42 °C) generated by BSe-Si disks exposed to 800 nm laser irradiation significantly reduced bacterial viability when combined with gentamicin, achieving a 55–58% reduction in vitro and a 51% reduction in vivo. However, this was less than a 1 log reduction and not clinically significant. Nonetheless, these findings align with previous studies, suggesting that elevated temperatures can enhance antibiotics by increasing biofilm permeability [45,46]. The reason for the minimal reduction in CFUs in vitro was most likely due to the low time of antibiotic exposure following treatment or the short time for laser exposure and heating. In the previous literature, heating times have only been a few minutes long at high temperatures, followed by 24 h of biofilms being incubated with antibiotics, which resulted in a significant loss of biofilm bacteria [93]. Alternatively, a few hours of mild hyperthermia at 45 °C plus antibiotics did not reduce bacteria significantly [94]. Our work is on par with the application of mild hyperthermia, when antibiotics are provided for a short time. Our goal was to quantify the effects of this newly developed PTT nanocomposite based on semiconducting polymer nanoparticles where the biofilm is adjacent to the heat-generating material during mild hyperthermia. Our results were good for *S. aureus*, although experimental parameters, especially prolonged time at elevated temperature, could have been altered to yield improved results.

A few factors that limited the success of the mild hyperthermia plus antibiotic utilization in vivo include not accounting for the cooler temperature of the tissue where the implant was placed. We assumed a start temperature of 37 °C for the skin of the mice, and that a ΔT = 5 °C would be sufficient for achieving mild hyperthermia, although the skin was most likely cooler (probably closer to 34 °C) [95]. Results did lead to lower bioluminescence, which indicates reduced metabolically active bacteria, or less bacteria. Since the laser treatment time, and therefore overall heating, was brief, at 25 s, this might have been one reason for the limited reduction in CFUs. Only one PTT treatment (compared to others in the literature where biofilm disruption took three cycles) was utilized, and only one bolus of antibiotics was given, which are variables that should be optimized in future evaluations using heat to improve antibiotic efficacy against biofilms [96]. Biofilms were allowed to recover for 24 h after PTT in the animal model. We may have seen a more dramatic impact if we had evaluated viable CFUs and bioluminescence immediately after treatment. However, this is clinically relevant since biofilm regrowth is a critical parameter affecting patient care.

Based on the fluorescence imaging analysis, hyperthermia appears to alter biofilm, which may break up biofilm to make it more susceptible to antibiotics. Heat also boosts bacterial metabolism, which can be used to augment antibiotics. For example, 10 min of 40 °C has been shown to boost bacterial metabolism [46]. Even if bacteria are not directly killed with high temperatures, if their metabolism is increased, this may benefit biofilm removal. Biofilm protein synthesis genes and heat shock protein genes can be altered with heat, complicating our understanding of how to best capitalize upon the synergy between antibiotics and heat [97,98,99,100,101,102]. Effects may be bacterial strain-dependent; hence, for clinical utilization, high heat may be best to ensure biofilm eradication. A previous study demonstrated higher synergy of antibiotics and heat in biofilms that had more eDNA, supporting the need to understand how temperatures alter biofilm composition [102]. While our application of PTT was safe, higher temperatures might be better. A potential limitation is that lasers at lower powers have reduced penetration depth and may only reach a few cm into tissue [68,70,103]. We envision PTT implants at deeper depths only in instances where there may be a catheter for delivery of the light via fiber optics (urinary catheters, central venous lines). Alternatively, light could be applied to regions where implant materials become infected at minimal depth or open access (silicone tubes for Eustachian tube disfunction or hemodialysis catheters) [7,104,105,106,107,108]. Similarly to oncologic applications, measuring hyperthermia in real time in vivo is a parameter that would help ensure optimal PTT for augmenting antibiotics [109,110,111,112]. Although real-time temperature feedback would be wonderful, we do not envision this as an easily accessible possibility. Magnetic resonance thermography is being developed but may not be practical for biofilms due to costs [113].

Our rationale for using two *S. aureus* strains that should produce different amounts of biomass was to observe any differences, although we had anticipated that the results might be more similar than what was observed. The results of the polysaccharide staining are perplexing since they are variable yet show a decrease in Xen 29 for PTT and only an increase when 5 W, 60 s was used with Xen 40. These trends were not expected and may be due to localized water content of the biofilms, which may vary with the different treatment regimens (more rapid water loss and biofilm compaction), which could lead to altered dye penetration [58]. More than likely, the results are statistical variance in the polysaccharide staining due to biofilm washing. It was not expected to have increased polysaccharides since the biofilms were stained immediately after PTT and not allowed to develop further. Therefore, the increased polysaccharide content for Xen 40 bacteria at 5 W, 60 s compared to the 1 W, 300 s or 3 W, 100 s may be crystallization of the polysaccharides with the more rapid temperature change. More characterization of the phenomenon is needed in future studies. The increase in eDNA is especially striking since the trend for Xen 40 increases with PTT is uniform for all laser regimes. Heat has previously been shown to decrease eDNA [114]. This result for Xen 40 makes sense compared to the data in Figure 6, where a greater reduction in biomass is observed for Xen 40 versus Xen 29. Thermal disruption of the biofilm kills bacteria, releasing eDNA. Data from Figure 7 also confirms the imaging data that there is more eDNA from Xen 40 compared to Xen 29, since eDNA can facilitate biofilm development, and there is more biofilm-on-biofilm regrowth for Xen 40. Since the imaging was performed immediately after PTT, genetic changes or heat shock stress are probably not causative factors.

Given the potential for rapid heating using ablative PTT, which should not allow development of heat resistance, this therapy should have broad-spectrum applicability. Although 600 J is often used for PTT, in this work, we demonstrated the efficacy of using a lower total fluence of 300 J utilizing 1 W, 300 s; 3 W, 100 s; or 5 W, 60 s. Different laser fluences (300 J) indicate that higher laser powers (3 W or 5 W) may be optimal, although the effects of higher powers on tissue must be balanced. Even though both 3 W and 5 W had approximately the same maximum temperature increase of 27 °C, 3 W applied for a longer time (100 s) had better outcomes for reducing both Xen 40 and Xen 29 biofilm bacteria. Even though both 3 W, 100 s and 5 W, 60 s generate the same temperature of about 27 °C, the prolonged time of 100 s may be key, since it was effective for both *S. aureus* strains. A major outcome of this work is the demonstration that PTT applied for either 60 s or 100 s is superior to bulk heating (using a water bath). This highlights the potential of having direct contact between biofilm and the heat source. While our work has proven that PTT might be better at reducing biofilm, safety considerations remain. Collateral tissue damage should be minimized since the high temperatures are only needed at the biofilm–implant interface. For example, inductive heating of metallic joint prosthesis has been shown to reduce biofilm, with those teams also supporting the benefit of heating the biofilm/implant interface to minimize collateral damage [93,115]. Although tissue damage can occur with our proposed technology, if the application of heat is brief, thermal transfer that leads to tissue damage will be minimal, allowing potential for recovery. Thermal ablation is routinely used in oncologic treatments, with some collateral tissue damage, but the benefits for eliminating both cancer and recalcitrant biofilms should outweigh any costs.

Of consideration is that high-temperature ablation may damage antibiotics; therefore, the prudent choice of temperature and the time of application into an infected implant area needs to be considered when used in conjunction with antibiotics. Another benefit of rapid PTT that leads to ablation of biofilms is that it minimizes the development of treatment resistance genes. PTT may aid in clinical biofilm removal since it is superior to bulk heating due to a more direct interface of thermal transfer from BSe-Si to bacteria and biofilm components. In this work, we propose that ablative PTT may be best to break the biofilm, with surrounding areas subjected to mild hyperthermia suitable for the augmentation of antibiotics, plus the potential for hyperthermia to augment a local immune response.

Xen 29 biofilms could be disrupted by BSe-Si-induced PTT; however, Xen 40 biofilms exhibited greater susceptibility to thermal stress, as evidenced by increased cell death, eDNA, and polysaccharide content following PTT. Fluorescence imaging revealed that PTT not only reduced bacterial viability but also altered the composition of the biofilm matrix, potentially impairing biofilm resilience and regrowth. This multifaceted disruption is particularly valuable given the limitation of antibiotics alone, which often fail to penetrate biofilms or eliminate persister cells [45,47,116,117]. The vast differences in polysaccharide expression remain perplexing and warrant an alternative metric using mass analysis of the alkali and water-soluble polysaccharides of *S. aureus* biofilms exposed to hyperthermia. Differences in *S. aureus* biofilms to hyperthermia are most likely due to polysaccharide and eDNA differences in biofilm composition and warrant further investigation into tailored therapeutic strategies. The key takeaway is that using PTT to kill bacteria and biofilms is not the only determining factor in successful treatment, since strain differences between bacteria, and hence differences in biofilm composition, can impact the acute biofilm viability but also the potential for biofilm regrowth. Our results confirm the benefits of PTT across multiple *S. aureus* strains and support the potential for future evaluation against other bacteria, including Gram-negative types.

Residual biomass can become supportive of biofilm regrowth, yet this has not been extensively considered in the previous literature [118]. Heat is an effective strategy for reducing *S. aureus* biomass; however, if not all bacteria are killed, biofilm regrowth can occur. Only a handful of papers have demonstrated hyperthermia on medical-grade materials, with one paper specifically identifying that heat can help stop biofilm attachment and development if used during the initial phases [89]. Our results demonstrate that biofilm regrowth was significantly inhibited following exposure to 64 °C for 100 s, suggesting that thermal treatment may offer durable suppression of biofilm. This effect was observed using bulk heating, and it will be interesting to discover if PTT confers the same advantage. Biomass must be eliminated to truly combat the clinical problem, especially if inflammatory components of residual biofilms induce fibrosis around an implant. There is also the potential for bacteria liberated from vivo biofilms following hyperthermia to trigger immune cell responses around the implant. As evidenced here, following hyperthermia, even elevated temperatures of 64 °C resulted in a 12 to 47% biomass rebound. This result is compelling for combining the synergistic benefits of hyperthermia with antimicrobial agents. Heat shock experiments corroborated the efficacy of thermal stress in compromising biofilm viability and structure yet highlight that PTT yields a greater reduction than bulk heating.

Reducing the bacterial burden around an implant can lead to better wound healing and more organized collagen, hence leading to the potential for less fibrosis around medical implants and a reduction in pain [119,120,121]. Heat can also disrupt viruses and fungi, which may be especially clinically relevant in combination with localized bacterial biofilm and is not often considered [122]. Having implanted materials that allow for in situ treatment eliminates the need for implant removal, thereby providing new clinical options to inhibit biofilm formation or disrupt established biofilms. Such a benefit has been shown with metal implants, although it is more difficult to generate heat in non-metallic, polymer-based materials. The clinical goal should be to eliminate the biofilm, to minimize biofilm remnants being re-colonized with bacteria, and biofilm regrowth. Killing persister cells which are metabolically dormant may be another key. Mild hyperthermia for longer times may be both safe and effective, although treatment time needs to be short to minimize heat shock resistance. While the current study provides compelling evidence for the efficacy of PTT using BSe-Si nanocomposites against *S. aureus* biofilms, several limitations should be acknowledged. This study used two *S. aureus* strains, which may not fully represent the diversity of clinical isolates or polymicrobial biofilms commonly encountered in device-associated infections. The in vivo model involved subcutaneous implantation in mice, which does not fully replicate the complexity of human device environments, and only one laser parameter was used in animals due to dermal sensitivity. Future work should evaluate the efficacy of BSe-Si PTT against polymicrobial biofilms and other clinically relevant pathogens, especially *Pseudomonas aeruginosa,* which often co-localizes with *S. aureus* in complex wounds. Long-term safety and biocompatibility of the nanocomposite also require further exploration before clinical translation. Finally, the feasibility of laser delivery to internal devices remains a challenge for clinical translation since the depth of light penetration is limited to less than 2 cm [68,123,124]. Further studies should also explore the integration of PTT with other antimicrobial agents, repeated cycles of PTT, and assess biofilm recurrence over extended timeframes.

## 5. Conclusions

Biofilm-associated infections on silicone-based medical devices pose a significant challenge in clinical settings due to poor antibiotic penetration and the persistence of resistant bacterial populations. This study demonstrates that PTT using PCPDTBSe nanoparticles offers a promising strategy for combating *S. aureus* biofilms on silicone-based medical devices. Mild hyperthermia with antibiotics significantly reduced bacterial viability, whereas ablative temperatures killed bacteria, inhibited biofilm regrowth, and disrupted biofilm composition. Overall, this work highlights the potential of nanocomposite-based PTT as a non-invasive, adjunctive strategy for managing biofilm-associated infections on medical devices. This approach may overcome key limitations of current therapies and reduce the need for device removal or surgical intervention, thus supporting the need for the further development of PTT, particularly in the context of device-associated infections, where there exists ample potential to modify heat generation through judicious choices of laser power, time of laser or heat exposure, and the type and concentration of NPs in the composite.

## Data Availability

The original contributions presented in this study are included in the article/Appendix A. Further inquiries can be directed to the corresponding author.

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
