# Peer review of "Semiconducting Polymer-Based Nanocomposite for Photothermal Elimination of Staphylococcus aureus Biofilm"

_microorganisms, 2025, doi:10.3390/microorganisms13112568_

Round 1
Reviewer 1 Report
Comments and Suggestions for Authors
In this work, Sanchez used polymer-based photothermal agent-modifed silicone (Si) medical devices to investigate the pure PTT and PTT/ antibiotic combined antibacterial/antibiofilm potential. Similar investigations have been done before. Therefore, it is not recommended to publish in the microorganisms. Following are some recommendations to improve the manuscript.
1. The polymer molecular structure should be offered in the manuscript.
2. The evidence to confirm polymer has been modified on silicone should be provided. and how many PTT agents are on the surface of the silicone?
3. The laser power is beyond the clinically permitted level.
4. The photothermal profile of pure photothermal agent-modified silicone (Si) and related controls should be provided
Author Response
In this work, Sanchez used polymer-based photothermal agent-modifed silicone (Si) medical devices to investigate the pure PTT and PTT/ antibiotic combined antibacterial/antibiofilm potential. Similar investigations have been done before. Therefore, it is not recommended to publish in the microorganisms. Following are some recommendations to improve the manuscript.
We thank the reviewer for taking the time to read our manuscript draft. We appreciate their effort and are distraught that we were unable to effectively convey the novelty of the work and why it should be published with Microorganisms. We have undertaken revisions to make the novelty in both the material composition, in addition to developing a deeper understanding of biofilm treated with heat, for which biofilm composition has not been evaluated before.
- The polymer molecular structure should be offered in the manuscript.
The chemical structure of PCPDTBSe has been included as Scheme 1 in the introduction section.
- The evidence to confirm polymer has been modified on silicone should be provided. and how many PTT agents are on the surface of the silicone?
We apologize, as there seems to be some confusion for the reviewer based on our text, which we have clarified in the methods section. PCPDTBSe nanoparticles were incorporated throughout silicone to obtain a homogeneous mixture before silicone curing. PCPDTBSe nanoparticles were not modified in any way. There was no precipitation of the nanoparticles. We do not know how many particles were located exactly at the surface as they were distributed throughout and we utilized the mass of nanoparticles to mass of silicone instead of using number of nanoparticles. We could make a mathematical calculation but feel this may not provide much value since material thickness may change with different iterations of the nanocomposite material. Given that the laser light will penetrate the entirety of the thickness of the BSe-Si disk for heat generation, quantifying the number of nanoparticles at the surface was not utilized. Our goal with developing the nanocomposite was to have a more durable material for handling in the lab, in lieu of a thin film coating on silicone, which we have examined previously. Developing the entire thickness of the disk to be nanocomposite eliminates the potential for thin film delamination. We have also revised the discussion section to emphasize the benefit of our approach.
- The laser power is beyond the clinically permitted level.
The laser used is an FDA-approved laser for dermal lesions and soft tissue pain management. Doses as high as 10 W can be safely applied for minutes at a time to human skin, and we utilize 5W routinely in our wound care clinic. We have included references to justify the laser power. Since mouse skin is much thinner, before we initiated the study, we tested 3W, 25s and 5W, 12s (these parameters generate the same amount of heat (about 5°C) from the BSe-Si disks) on the skin of mice without the inclusion of BSe-Si disks. We observed dermal damage with 5W, 12s, but not with 3W, 25s. Hence, we used 3W for treating the mice with BSe-Si disks and antibiotic.
- The photothermal profile of pure photothermal agent-modified silicone (Si) and related controls should be provided
We apologize as there seems to be some confusion. We refer the reviewer to Figure 4 which includes both unmodified silicone and the nanocomposite, with temperature changes over time using 1, 3, or 5W of 800nm light. Figure 1also has all appropriate controls of Si versus BSe-Si disks with the 3W or 5W of laser power that was used in the animal model.
Reviewer 2 Report
Comments and Suggestions for Authors
Minor comments
(L9–27) The abstract reports significant antibacterial reductions through photothermal therapy (PTT) using PCPDTBSe-based nanocomposites. Could the authors further delineate how this nanocomposite system advances beyond previously established PTT platforms for Staphylococcus aureus biofilms, particularly in terms of photothermal conversion efficiency and material integration?
(L31–46) While the introduction effectively highlights the burden of healthcare-associated infections, the direct correlation between this clinical challenge and the proposed silicone–nanocomposite photothermal approach requires further justification. How does the present system address specific limitations of existing anti-biofilm device coatings?
(L75–91) The study distinguishes between mild and ablative hyperthermia regimes but provides limited mechanistic insight. Could the authors elaborate on the specific cellular and biochemical pathways by which each thermal regime exerts bactericidal effects on S. aureus biofilms?
(L110–132) The synthesis of PCPDTBSe nanoparticles involves multi-step processing and solvent exchange. Were the particle size distribution, morphology, and surface stability systematically characterized to ensure reproducibility and uniform photothermal behavior across different experimental batches?
(L133–148) Temperature calibration was conducted using fiberoptic probes to achieve a target ΔT of 5 °C. How was spatial temperature uniformity across the disk surface validated, and was any form of infrared thermography or thermal modeling employed to confirm homogeneity during laser irradiation?
(L150–159) Two bioluminescent S. aureus strains (Xen29 and Xen40) were selected for biofilm studies. Please clarify the scientific rationale behind this selection and discuss how the differences in extracellular polymeric substance (EPS) composition or biofilm maturity might influence the photothermal response.
(L206–239) The in vivo implantation study employs subcutaneous disks in murine models. Were histopathological evaluations conducted to assess potential dermal or subdermal thermal injury, and how were laser parameters optimized to balance antibacterial efficacy with host tissue safety?
(L321–338) The mild-hyperthermia treatment achieved approximately 55–58 % bacterial reduction, which corresponds to less than a one-log CFU decrease. How do the authors interpret this modest reduction in terms of biological relevance, and what additional evidence could strengthen its clinical significance?
(L369–392) The in vivo data demonstrate a 51 % reduction in CFUs following combined PTT and antibiotic therapy. Could the authors provide statistical validation with replicate analyses or discuss whether longer treatment durations and repeated exposures might yield clinically meaningful outcomes?
(L404–424) The ablative hyperthermia experiments indicate high ΔT values and efficient bacterial elimination. However, elevated temperatures could pose risks of collateral tissue damage. What thermal safety thresholds were considered, and how might laser fluence or exposure time be optimized for translational applications?
(L551–606) The fluorescence imaging results reveal increased extracellular DNA and polysaccharide levels following PTT, particularly in the Xen40 strain. Could the authors propose mechanistic explanations for these compositional alterations—such as heat-induced matrix remodeling or stress-mediated release of nucleic acids—and discuss their implications for biofilm resilience or regrowth?
(L613–674) The discussion outlines the potential of PTT as an adjunct to antibiotic therapy for medical device-associated infections. What specific engineering or clinical challenges—such as laser penetration depth, real-time temperature feedback, and safe in situ device heating—must be addressed to enable practical clinical translation of this nanocomposite system?
Round 2
Reviewer 1 Report
Comments and Suggestions for Authors
Accept in present form